# Neural Spatio-Temporal Point Processes

**Ricky T. Q. Chen**[*]
University of Toronto; Vector Institute
rtqichen@cs.toronto.edu

**Brandon Amos, Maximilian Nickel**
Facebook AI Research
{bda,maxn}@fb.com

## Abstract

We propose a new class of parameterizations for spatio-temporal point processes which leverage Neural ODEs as a computational method and enable flexible, high-fidelity models of discrete events that are localized in continuous time and space. Central to our approach is a combination of continuous-time neural networks with two novel neural architectures, *i.e.*, Jump and Attentive Continuous-time Normalizing Flows. This approach allows us to learn complex distributions for both the spatial and temporal domain and to condition non-trivially on the observed event history. We validate our models on data sets from a wide variety of contexts such as seismology, epidemiology, urban mobility, and neuroscience.

## 1 Introduction

Modeling discrete events that are localized in continuous time and space is an important task in many scientific fields and applications. Spatio-temporal point processes (STPPs) are a versatile and principled framework for modeling such event data and have, consequently, found many applications in a diverse range of fields. This includes, for instance, modeling earthquakes and aftershocks (Ogata, 1988; 1998), the occurrence and propagation of wildfires (Hering et al., 2009), epidemics and infectious diseases (Meyer et al., 2012; Schoenberg et al., 2019), urban mobility (Du et al., 2016), the spread of invasive species (Balderama et al., 2012), and brain activity (Tagliazucchi et al., 2012).

It is of great interest in all of these areas to learn high-fidelity models which can jointly capture spatial and temporal dependencies and their propagation effects. However, existing parameterizations of STPPs are strongly restricted in this regard due to computational considerations: In its general form, STPPs require solving multivariate integrals for computing likelihood values and thus have primarily been studied within the context of different approximations and model restrictions. This includes, for instance, restricting the model class to parameterizations with known closed-form solutions (*e.g.*, exponential Hawkes processes (Ozaki, 1979)), to restrict dependencies between the spatial and temporal domain (*e.g.*, independent and unpredictable marks (Daley & Vere-Jones, 2003)), or to discretize continuous time and space (Ogata, 1998). These restrictions and approximations—which can lead to mis-specified models and loss of information—motivated the development of neural temporal point processes such as Neural Hawkes Processes (Mei & Eisner, 2017) and Neural Jump SDEs (Jia & Benson, 2019). While these methods are more flexible, they can still require approximations such as Monte-Carlo sampling of the likelihood (Mei & Eisner, 2017; Nickel & Le, 2020) and, most importantly, only model restricted spatial distributions (Jia & Benson, 2019).

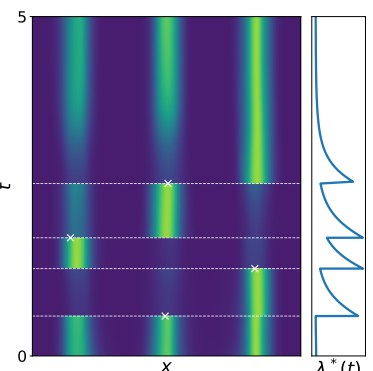

Figure 1: Color is used to denote $p(x|t)$, which can be evaluated for Neural STPPs. After observing an event in one mode, the model is instantaneously updated as it strongly expects an event in the next mode. After a period of no observations, the model smoothly reverts back to the marginal distribution.

To overcome these issues, we propose a new class of parameterizations for spatio-temporal point processes which leverage Neural ODEs as a computational method and allows us to define flexible,

---

[*]Work done while at Facebook AI Research.

high-fidelity models for spatio-temporal event data. We build upon ideas of Neural Jump SDEs (Jia & Benson, 2019) and Continuous-time Normalizing Flows (CNFs; Chen et al. 2018; Grathwohl et al. 2019; Mathieu & Nickel 2020) to learn parametric models of spatial (or mark[1]) distributions that are defined continuously in time. Normalizing flows are known to be flexible universal density estimators (*e.g.* Huang et al. 2018; 2020; Teshima et al. 2020; Kong & Chaudhuri 2020) while retaining computational tractability. As such, our approach allows the computation of exact likelihood values even for highly complex spatio-temporal distributions, and our models create smoothly changing spatial distributions that naturally benefits spatio-temporal modeling. Central to our approach, are two novel neural architectures based on CNFs—using either discontinuous jumps in distribution or self-attention—to condition spatial distributions on the event history. To the best of our knowledge, this is the first method that combines the flexibility of neural TPPs with the ability to learn high-fidelity models of continuous marks that can have complex dependencies on the event history. In addition to our modeling contributions, we also construct five new pre-processed data sets for benchmarking spatio-temporal event models.

## 2 BACKGROUND

In the following, we give a brief overview of two core frameworks which our method builds upon, *i.e.*, spatio-temporal point processes and continuous-time normalizing flows.

**Event Modeling with Point Processes**   Spatio-temporal point processes are concerned with modeling sequences of random events in continuous space and time (Moller & Waagepetersen, 2003; Baddeley et al., 2007). Let $\mathcal{H} = \{(t_i, \boldsymbol{x}_i)\}_{i=1}^n$ denote the sequence of event times $t_i \in \mathbb{R}$ and their associated locations $\boldsymbol{x}_i \in \mathbb{R}^d$, the number of events $n$ being also random. Additionally, let $\mathcal{H}_t = \{(t_i, \boldsymbol{x}_i) \mid t_i < t, t_i \in \mathcal{H}\}$ denote the history of events predating time $t$. A spatio-temporal point process is then fully characterized by its *conditional intensity function*

$$\lambda(t, \boldsymbol{x} \mid \mathcal{H}_t) \triangleq \lim_{\Delta t \downarrow 0, \Delta \boldsymbol{x} \downarrow 0} \frac{\mathbb{P}\left(t_i \in [t, t+\Delta t], \boldsymbol{x}_i \in B(\boldsymbol{x}, \Delta \boldsymbol{x}) \mid \mathcal{H}_t\right)}{|B(\boldsymbol{x}, \Delta \boldsymbol{x})|\Delta t} \ . \tag{1}$$

where $B(\boldsymbol{x}, \Delta \boldsymbol{x})$ denotes a ball centered at $\boldsymbol{x} \in \mathbb{R}^d$ and with radius $\Delta \boldsymbol{x}$. The only condition is that $\lambda(t, \boldsymbol{x} \mid \mathcal{H}_t) \geq 0$ and need not be normalized. Given $i-1$ previous events, the conditional intensity function describes therefore the instantaneous probability of the $i$-th event occurring at $t$ and location $\boldsymbol{x}$. In the following, we will use the common star superscript shorthand $\lambda^*(t, \boldsymbol{x}) = \lambda(t, \boldsymbol{x} \mid \mathcal{H}_t)$ to denote conditional dependence on the history. The joint log-likelihood of observing $\mathcal{H}$ within a time interval of $[0, T]$ is then given by (Daley & Vere-Jones, 2003, Proposition 7.3.III)

$$\log p(\mathcal{H}) = \sum_{i=1}^n \log \lambda^*(t_i, \boldsymbol{x}_i) - \int_0^T \int_{\mathbb{R}^d} \lambda^*(\tau, \boldsymbol{x}) \ d\boldsymbol{x} d\tau. \tag{2}$$

Training general STPPs with maximum likelihood is difficult as eq. (2) requires solving a multivariate integral. This need to compute integrals has driven research to focus around the use of kernel density estimators (KDE) with exponential kernels that have known anti-derivatives (Reinhart et al., 2018).

**Continuous-time Normalizing Flows**   Normalizing flows (Dinh et al., 2014; 2016; Rezende & Mohamed, 2015) is a class of density models that describe flexible distributions by parameterizing an invertible transformation from a simpler base distribution, which enables exact computation of the probability of the transformed distribution, without any unknown normalization constants.

Given a random variable $\boldsymbol{x}_0$ with known distribution $p(\boldsymbol{x}_0)$ and an invertible transformation $F(x)$, the transformed variable $F(\boldsymbol{x}_0)$ is a random variable with a probability distribution function that satisfies

$$\log p(F(\boldsymbol{x}_0)) = \log p(\boldsymbol{x}_0) - \log \left| \det \frac{\partial F}{\partial x}(\boldsymbol{x}_0) \right|. \tag{3}$$

There have been many advances in parameterizing $F$ with flexible neural networks that also allow for cheap evaluations of eq. (3). We focus our attention on Continuous-time Normalizing Flows (CNFs), which parameterizes this transformation with a Neural ODE (Chen et al., 2018). CNFs

---

[1]We regard any marked temporal point process with continuous marks as a spatio-temporal point process.

define an infinite set of distributions on the real line that vary smoothly across time, and will be our core component for modeling events in the spatial domain.

Let $p(\boldsymbol{x}_0)$ be the base distribution[2]. We then parameterize an instantaneous change in the form of an ordinary differential equation (ODE), $\frac{d\boldsymbol{x}_t}{dt} = f(t, \boldsymbol{x}_t)$, where the subscript denotes dependence on $t$. This function can be parameterized using any Lipschitz-continuous neural network. Conditioned on a sample $\boldsymbol{x}_0$ from the base distribution, let $\boldsymbol{x}_t$ be the solution of the initial value problem [3] at time $t$, *i.e.* it is from a trajectory that passes through $\boldsymbol{x}_0$ at time 0 and satisfies the ODE $d\boldsymbol{x}_t/dt = f$. We can express the value of the solution at time $t$ as

$$\boldsymbol{x}_t = \boldsymbol{x}_0 + \int_0^t f(t, \boldsymbol{x}_\tau)d\tau. \tag{4}$$

The distribution of $\boldsymbol{x}_t$ then also continuously changes in $t$ through the following equation,

$$\log p(\boldsymbol{x}_t|t) = \log p(\boldsymbol{x}_0) - \int_0^t \mathrm{tr}\left(\frac{\partial f}{\partial x}(\tau, \boldsymbol{x}_\tau)\right) \, d\tau. \tag{5}$$

In practice, eq. (4) and eq. (5) are solved together from 0 to $t$, as eq. (5) alone is not an ordinary differential equation but the combination of $\boldsymbol{x}_t$ and $\log p(\boldsymbol{x}_t)$ is. The trace of the Jacobian $\frac{\partial f}{\partial x}(\tau, \boldsymbol{x}_\tau)$ can be estimated using a Monte Carlo estimate of the identity (Skilling, 1989; Hutchinson, 1990), $\mathrm{tr}(A) = \mathbb{E}_{v \sim \mathcal{N}(0,1)}[v^\mathsf{T} A v]$. This estimator relies only on a vector-Jacobian product, which can be efficiently computed in modern automatic differentiation and deep learning frameworks. This has been used (Grathwohl et al., 2019) to scale CNFs to higher dimensions using a Monte Carlo estimate of the log likelihood objective,

$$\log p(\boldsymbol{x}_t|t) = \log p(\boldsymbol{x}_0) - \mathbb{E}_{v \sim \mathcal{N}(0,1)}\left[\int_0^t v^\mathsf{T} \frac{\partial f}{\partial x}(\tau, \boldsymbol{x}_\tau)v \, d\tau\right], \tag{6}$$

which, even if only one sample of $v$ is used, is still amenable to training with stochastic gradient descent. Gradients with respect to any parameters in $f$ can be computed with constant memory by solving an adjoint ODE in reverse-time as described in Chen et al. (2018).

## 3  NEURAL SPATIO-TEMPORAL POINT PROCESSES

We are interested in modeling high-fidelity distributions in continuous time and space that can be updated based on new event information. For this purpose, we use the Neural ODE framework to parameterize a STPP by combining ideas from Neural Jump SDEs and Continuous Normalizing Flows to create highly flexible models that still allow exact likelihood computation.

We first (re-)introduce necessary notation. Let $\mathcal{H} = \{(t_i, \boldsymbol{x}_{t_i}^{(i)})\}$ denote a sequence of event times $t_i \in [0, T]$ and locations $\boldsymbol{x}_{t_i}^{(i)} \in \mathbb{R}^d$. The superscript indicates an association with the $i$-th event, and the use of subscripting with $t_i$ will be useful later in the continuous-time modeling framework. Following Daley & Vere-Jones (2003), we decompose the conditional intensity function as

$$\lambda^*(t, \boldsymbol{x}) = \lambda^*(t) \, p^*(\boldsymbol{x} \mid t) \tag{7}$$

where $\lambda^*(t)$ is the *ground intensity* of the temporal process and where $p^*(\boldsymbol{x} \mid t)$ is the *conditional density* of a mark $\boldsymbol{x}$ at $t$ given $\mathcal{H}_t$. The star superscript is used as again shorthand to denote dependence on the history. Since $\int_{\mathbb{R}^d} p^*(\boldsymbol{x} \mid t) = 1$, eq. (7) allows us now to simplify the log-likelihood function of the joint process from eq. (2), such that

$$\log p(\mathcal{H}) = \underbrace{\sum_{i=1}^n \log \lambda^*(t_i) - \int_0^T \lambda^*(\tau) \, d\tau}_{\text{temporal log-likelihood}} + \underbrace{\sum_{i=1}^n \log p^*(\boldsymbol{x}_{t_i}^{(i)}|t_i)}_{\text{spatial log-likelihood}} \tag{8}$$

---

[2]This can be any distribution that is easy to sample from and evaluate log-likelihood for. We use the typical choice of a standard Normal in our experiments.

[3]This initial value problem has a unique solution if $f$ is Lipschitz-continuous along the trajectory to $\boldsymbol{x}_t$, by the Picard–Lindelöf theorem.

Furthermore, based on eq. (7), we can derive separate models for the ground intensity and conditional mark density which will be jointly conditioned on a continuous-time hidden state with jumps. In the following, we will first describe how we construct a latent dynamics model, which we use to compute the ground intensity $\lambda^*(t)$. We will then propose three novel CNF-based approaches for modeling the conditional mark density $p^*(\boldsymbol{x}|t)$. We will first describe an unconditional model, which is already a strong baseline when spatial event distributions only follow temporal patterns and there is little to no correlation between the spatial observations. We then devise two new methods of conditioning on the event history $\mathcal{H}$: one explicitly modeling instantaneous changes in distribution, and another that uses an attention mechanism which is more amenable to parallelism.

**Latent Dynamics and Ground Intensity**    For the temporal variables $\{t_i\}$, parameterize the intensity function using hidden state dynamics with jumps, similar to the work of Jia & Benson (2019). Specifically, we evolve a continuous-time hidden state $\boldsymbol{h}$ and set

$$\lambda^*(t) = g_\lambda(\boldsymbol{h}_t) \qquad \text{(Ground intensity)} \tag{9}$$

where $g_\lambda$ is a neural network with a softplus nonlinearity applied to the output, to ensure the intensity is positive. We then capture conditional dependencies through the use of a continuously changing state $\boldsymbol{h}_t$ with instantaneous updates when conditioned on an event.

The architecture is analogous to a recurrent neural network with a continuous-time hidden state (Mei & Eisner, 2017; Che et al., 2018; Rubanova et al., 2019) modeled by a Neural ODE. This provides us with a vector representation $\boldsymbol{h}_t$ at every time value $t$ that acts as both a summary of the history of events and as a predictor of future behavior. Instantaneous updates to $\boldsymbol{h}_t$ allow to incorporate abrupt changes to the hidden state that are triggered by observed events. This mechanism is important for modeling point processes and allows past events to influence future dynamics in a discontinuous way (*e.g.*, modeling immediate shocks to a system).

We use $f_h$ to model the continuous change in the form of an ODE and $g_h$ to model instantaneous changes based on an observed event.

$$\boldsymbol{h}_{t_0} = \boldsymbol{h}_0 \qquad \text{(An initial hidden state)} \tag{10}$$

$$\frac{d\boldsymbol{h}_t}{dt} = f_h(t, \boldsymbol{h}_t) \qquad \text{between event times} \qquad \text{(Continuous evolution)} \tag{11}$$

$$\lim_{\varepsilon \to 0} \boldsymbol{h}_{t_i+\varepsilon} = g_h\left(t_i, \boldsymbol{h}_{t_i}, \boldsymbol{x}_{t_i}^{(i)}\right) \qquad \text{at event times } t_i \qquad \text{(Instantaneous updates)} \tag{12}$$

The use of $\varepsilon$ is to portray that $\boldsymbol{h}_t$ is a *càglàd* function, *i.e.* left-continuous with right limits, with a discontinuous jump modeled by $g_h$.

The parameterization of continuous-time hidden states in the form of eqs. (10) to (12) has been used for time series modeling (Rubanova et al., 2019; De Brouwer et al., 2019) as well as TPPs (Jia & Benson, 2019). We parameterize $f_h$ as a standard multi-layer fully connected neural network, and use the GRU update (Cho et al., 2014) to parameterize $g_h$, as was done in Rubanova et al. (2019).

**Time-varying CNF**    The first model we consider is a straightforward application of the CNF to time-variable observations. Assuming that the spatial distribution is independent of prior events,

$$\log p^*(\boldsymbol{x}_{t_i}^{(i)}|t_i) = \log p(\boldsymbol{x}_{t_i}^{(i)}|t_i) = \log p(\boldsymbol{x}_0^{(i)}) - \int_0^{t_i} \operatorname{tr}\left(\frac{\partial f}{\partial x}(\tau, \boldsymbol{x}_\tau^{(i)})\right) d\tau \tag{13}$$

where $\boldsymbol{x}_\tau^{(i)}$ is the solution of the ODE $f$ with initial value $\boldsymbol{x}_{t_i}^{(i)}$, the observed event location, at $\tau = t_i$, the observed event time. The spatial distribution of an event modeled by a Time-varying CNF changes with respect to the time it occurs. Some spatio-temporal data sets exhibit mostly temporal patterns and little to no dependence on previous events in the spatial domain, which would make a time-varying CNF a good fit. Nevertheless, this model lacks the ability to capture spatial propagation effects, as it does not condition on previous event observations.

A major benefit of this model is the ability to evaluate the joint log-likelihood fully in parallel across events, since there are no dependencies between events. Most modern ODE solvers that we are aware of only allow a scalar terminal time. Thus, to solve all $n$ integrals in eq. (13) with a single call to an ODE solver, we can simply reparameterize all integrals with a consistent dummy variable and track

the terminal time in the state (see Appendix F for detailed explanation). Intuitively, the idea is that we can reparameterize ODEs that are on $t \in [0, t_i]$ into an ODE on $s \in [0, 1]$ using the change of variables $s = t/t_i$ (or $t = st_i$) and scaling the output of $f$ by $t_i$. The joint ODE is then

$$\frac{d}{ds} \underbrace{\begin{bmatrix} \boldsymbol{x}_s^{(1)} \\ \vdots \\ \boldsymbol{x}_s^{(n)} \end{bmatrix}}_{A_s} = \underbrace{\begin{bmatrix} t_1 f(st_1, \boldsymbol{x}_s^{(1)}) \\ \vdots \\ t_n f(st_n, \boldsymbol{x}_s^{(n)}) \end{bmatrix}}_{f(s, A_s)} \quad \text{which gives} \quad \underbrace{\begin{bmatrix} \boldsymbol{x}_0^{(1)} \\ \vdots \\ \boldsymbol{x}_0^{(n)} \end{bmatrix}}_{A_0} + \int_0^1 f(s, A_s) ds = \underbrace{\begin{bmatrix} \boldsymbol{x}_{t_1}^{(1)} \\ \vdots \\ \boldsymbol{x}_{t_n}^{(n)} \end{bmatrix}}_{A_1}. \quad (14)$$

Thus the full trajectories between $0$ to $t_i$ for all events can be computed in parallel using this augmented ODE by simply integrating once from $s = 0$ to $s = 1$.

**Jump CNF**  For the second model, we condition the dynamics defining the continuous normalizing flow on the hidden state $\boldsymbol{h}$, allowing the normalizing flow to update its distribution based on changes in $\mathcal{H}$. For this purpose, we define continuous-time spatial distributions by making again use of two components: (i) a continuous-time normalizing flow that evolves the distribution continuously, and (ii) a standard (discrete-time) flow model that changes the distribution instantaneously after conditioning on new events. As normalizing flows parameterize distributions through transformations of the samples, these continuous- and discrete-time transformations are composable in a straightforward manner and are end-to-end differentiable.

The generative process *of a single event* in a Jump CNF is given by:

$$\boldsymbol{x}_0 \sim p(\boldsymbol{x}_0) \qquad \text{(An initial distribution)} \qquad (15)$$

$$\frac{d\boldsymbol{x}_t}{dt} = f_x(t, \boldsymbol{x}_t, \boldsymbol{h}_t) \qquad \text{between event times} \qquad \text{(Continuous evolution)} \qquad (16)$$

$$\lim_{\varepsilon \to 0} \boldsymbol{x}_{t_i + \varepsilon} = g_x(t_i, \boldsymbol{x}_{t_i}, \boldsymbol{h}_{t_i}) \qquad \text{at event times } t_i \qquad \text{(Instantaneous updates)} \qquad (17)$$

The initial distribution can be parameterized by a normalizing flow. In practice, we set a base distribution at a negative time value and model $p(\boldsymbol{x}_0)$ using the same CNF parameterized by $f_x$. The instantaneous updates (or jumps) describe conditional updates in distribution after each new event has been observed. This conditioning on $\boldsymbol{h}_{t_i}$ is required for the continuous and instantaneous updates to depend on the history of observations. Otherwise, a Jump CNF would only be able to model the marginal distribution and behave similarly to a time-varying CNF. We solve for $\boldsymbol{h}_t$ alongside $\boldsymbol{x}_t$.

The final probability of an event $\boldsymbol{x}_t$ at some $t > t_n$ after observing $n$ events is given by the sum of changes according to the continuous- and discrete-time normalizing flows.

$$\log p^*(\boldsymbol{x}_t | t) = \log p(\boldsymbol{x}_0)$$
$$+ \sum_{t_i \in \mathcal{H}_t} \underbrace{\left( - \int_{t_{i-1}}^{t_i} \mathrm{tr}\left( \frac{\partial f(\tau, \boldsymbol{x}_\tau, \boldsymbol{h}_\tau)}{\partial x} \right) d\tau - \log \left| \det \frac{\partial g_x(t_i, \boldsymbol{x}_{t_i}, \boldsymbol{h}_{t_i})}{\partial x} \right| \right)}_{\text{Change in density up to last event}}$$
$$+ \underbrace{\int_{t_n}^{t} -\mathrm{tr}\left( \frac{\partial f(\tau, \boldsymbol{x}_\tau, \boldsymbol{h}_\tau)}{\partial x} \right) d\tau}_{\text{Change in density from last event to } t} \quad (18)$$

As the instantaneous updates must be applied sequentially in a Jump CNF, we can only compute the integrals in eq. (18) one at a time. As such, the number of initial value problems scales linearly with the number of events in the history because the ODE solver must be restarted between each instantaneous update to account for the discontinuous change to state. This incurs a substantial cost when the number of events is large.

**Attentive CNF**  To design a spatial model with conditional dependencies that alleviates the computational issues of Jump CNFs and can be computed in parallel, we make use of efficient attention mechanisms based on the Transformer architecture (Vaswani et al., 2017). Denoting only the spatial variables for simplicity, each conditional distribution $\log p(\boldsymbol{x}_{t_i} \mid \mathcal{H}_{t_i})$ can be modeled by a CNF that

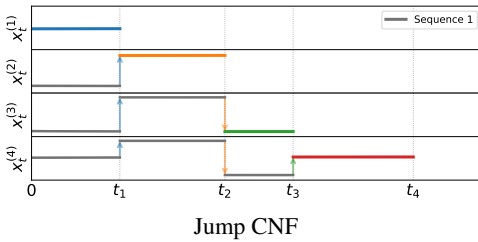 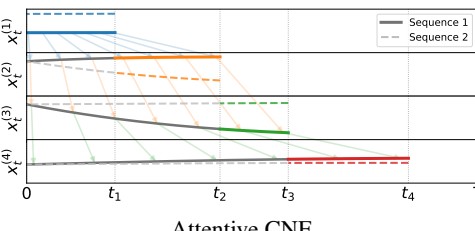

Figure 2: Visualization of the sampling paths of Neural STPP models for a 1-D spatio-temporal data set where $\{t_i\}_{i=1}^4$ are event times. The Jump CNF uses instantaneous jumps to update its distribution based on newly observed events while the Attentive CNF depends continuously on the sampling paths of prior events. We additionally visualize a second sequence for the Attentive CNF where the random base samples $\{x_0^{(i)}\}_{i=2}^4$ are the same as in sequence 1. Even so, the sampling paths are different because the first event is different, effectively leading to different conditional spatial distributions. See Figure 9 for visualizations of the learned density.

depends on the sample path of prior events. Specifically, we take the dummy-variable reparameterization of eq. (14) and modify it so that the $i$-th event depends on all previous events using a Transformer architecture for $f$,

$$\frac{d}{ds}\begin{bmatrix}\boldsymbol{x}_s^{(1)}\\\boldsymbol{x}_s^{(2)}\\\vdots\\\boldsymbol{x}_s^{(n)}\end{bmatrix}=\begin{bmatrix}t_1 f\left(st_1,\{\boldsymbol{x}_s^{(i)}\}_{i=1}^1,\{\boldsymbol{h}_{t_i}\}_{i=1}^1\right)\\t_2 f\left(st_2,\{\boldsymbol{x}_s^{(i)}\}_{i=1}^2,\{\boldsymbol{h}_{t_i}\}_{i=1}^2\right)\\\vdots\\t_n f\left(st_n,\{\boldsymbol{x}_s^{(i)}\}_{i=1}^n,\{\boldsymbol{h}_{t_i}\}_{i=1}^n\right)\end{bmatrix}:=f_{\text{Attn}}.\tag{19}$$

With this formulation, the trajectory of $\boldsymbol{x}_\tau^{(i)}$ depends continuously on the trajectory of $\boldsymbol{x}_\tau^{(j)}$ for all $j < i$ and the hidden states $\boldsymbol{h}$ prior to the $i$-th event. Similar to eq. (14), an Attention CNF can now solve for the trajectories of all events in parallel while simultaneously depending non-trivially on $\mathcal{H}$.

To parameterize $f_{\text{Attn}}$, we use an embedding layer followed by two multihead attention (MHA) blocks and an output layer to map back into the input space. We use the Lipschitz-continuous multihead attention from Kim et al. (2020) as they recently showed that the dot product multihead attention (Vaswani et al., 2017) is not Lipschitz-continuous and thus may be ill-suited for parameterizing ODEs.

*Low-variance Log-likelihood Estimation* The variance of the Hutchinson stochastic trace estimator in eq. (6) grows with the squared Frobenius norm of the Jacobian, $\sum_{ij}\left[\partial f/\partial x\right]_{ij}^2$ (Hutchinson, 1990). For attentive CNFs, we can remove some of the non-diagonal elements of the Jacobian and achieve a lower variance estimator. The attention mechanism creates a block-triangular Jacobian, where each block corresponds to one event, but the elements outside of the block-diagonal are solely due to the multihead attention. By *detaching the gradient connections* between different events in the MHA blocks, we can create a surrogate

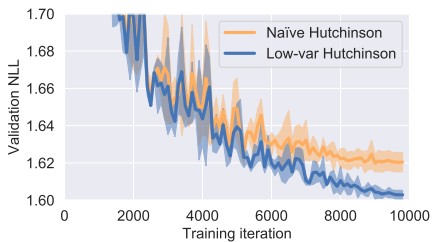

Figure 3: Lower variance estimates of the log-likelihood allows training better Attentive CNFs.

Jacobian matrix that do not contain cross-event partial derivatives. This effectively allows us to apply Hutchinson's estimator on a matrix that has the same diagonal elements as the Jacobian $\partial f/\partial x$—and thus has the same expected value—but has zeros outside of the block-diagonal, leading to a lower variance trace estimator. The procedure consists of selectively removing partial derivatives and is straightforward but notationally cumbersome; the interested reader can find the details in Appendix E.

This is similar in spirit to Chen & Duvenaud (2019) but instead of constructing a neural network that specifically allows cheap removal of partial derivatives, we make use of the fact that multihead attention already allows cheap removal of (cross-event) partial derivatives.

An ablation experiment is shown in Figure 3 for training on the PINWHEEL data set, where the lower variance estimates (and gradients) ultimate led to faster convergence and better converged models.

## 4 RELATED WORK

**Neural Temporal Point Processes**  Modeling real-world data using restricted models such as Exponential Hawkes Processes (Ozaki, 1979) may lead to poor results due to model mis-specification. While this has led to many works on improving the Hawkes process (*e.g.* Linderman & Adams 2014; Li & Zha 2014; Zhao et al. 2015; Farajtabar et al. 2017; Li & Ke 2020; Nickel & Le 2020), recent works have begun to explore neural network parameterizations of TPPs. A common approach is to use recurrent neural networks to accumulate the event history in a latent state from which the intensity value can then be derived. Models of this form include, for instance, Recurrent Marked Temporal Point Processes (RMTPPs; Du et al. 2016) and Neural Hawkes Processes (NHPs; Mei & Eisner 2017). In contrast to our approach, these methods can not compute the exact likelihood of the model and have to resort to Monte-Carlo sampling for its approximation. However, this approach is especially problematic for commonly occurring clustered and bursty event sequences as it either requires a very high sampling rate or ignores important temporal dependencies (Nickel & Le, 2020). To overcome this issue, Jia & Benson (2019) proposed Neural Jump SDEs which extend the Neural ODE framework and allow to compute the exact likelihood for neural TPPs, up to numerical errors. This method is closely related to our approach and we build on its ideas to compute the ground intensity of the STPP. However, current Neural Jump SDEs —as well as NHPs and RMTPPs—are not well-suited for modeling complex *continuous* mark distributions as they are restricted to methods such as Gaussian mixture models in the spatial domain. Finally, Shchur et al. (2019) and Mehrasa et al. (2019) considered combining TPPs with flexible likelihood-based models, however for different purposes as in our case, *i.e.*, for intensity-free learning of only temporal point processes.

**Continuous Normalizing Flows**  The ability to describe an infinite number of distributions with a Continuous Normalizing Flow has been used by a few recent works. Some works in computer graphics have used the interpolation effect of CNFs to model transformations of point clouds (Yang et al., 2019; Rempe et al., 2020; Li et al., 2020). CNFs have also been used in sequential latent variable models (Deng et al., 2020; Rempe et al., 2020). However, such works do not align the "time" axis of the CNF with the temporal axis of observations, and do not train on observations at more than one value of "time" in the CNF. In contrast, we align the time axis of the CNF with the time of the observations, directly using its ability to model distributions on a real-valued axis. A closely related application of CNFs to spatio-temporal data was done by Tong et al. (2020), who modeled the distribution of cells in a developing human embryo system at five fixed time values. In contrast to this, we extend to applications where observations are made at arbitrary time values, jointly modeling space and time within the spatio-temporal point process framework. Furthermore, Mathieu & Nickel (2020); Lou et al. (2020) recently proposed extensions of CNFs to Riemannian manifolds. For our proposed approach, this is especially interesting in the context of earth and climate science, as it allows us to model STPPs on the sphere simply by replacing the CNF with its Riemannian equivalent.

## 5 EXPERIMENTS

**Data Sets**  Many collected data can be represented within the framework of spatio-temporal events. We pre-process data from open sources and make them suitable for spatio-temporal event modeling. Each sequence in these data sets can contain up to thousands of variables, all the while having a large variance in sequence lengths. Varying across a wide range of domains, the data sets we consider are: earthquakes, pandemic spread, consumer demand for a bike sharing app, and high-amplitude brain signals from fMRI scans. We briefly describe these data sets here; further details, pre-processing steps, and data set diagnostics can be found in Appendix C. Code for preprocessing and training are open sourced at `https://github.com/facebookresearch/neural_stpp`.

PINWHEEL This is a synthetic data set with multimodal and non-Gaussian spatial distributions designed to test the ability to capture drastic changes due to event history (see fig. 5). The data set consists of 10 clusters which form a pinwheel structure. Events are sampled from a multivariate Hawkes process such that events from one cluster will increase the probability of observing events in the next cluster in a clock-wise rotation. Number of events per sequences ranges between 4 to 108.

EARTHQUAKES For modeling earthquakes and aftershocks, we gathered location and time of all earthquakes in Japan from 1990 to 2020 with magnitude of at least 2.5 from the U.S. Geological Survey (2020). Number of events per sequences ranges between 18 to 543.

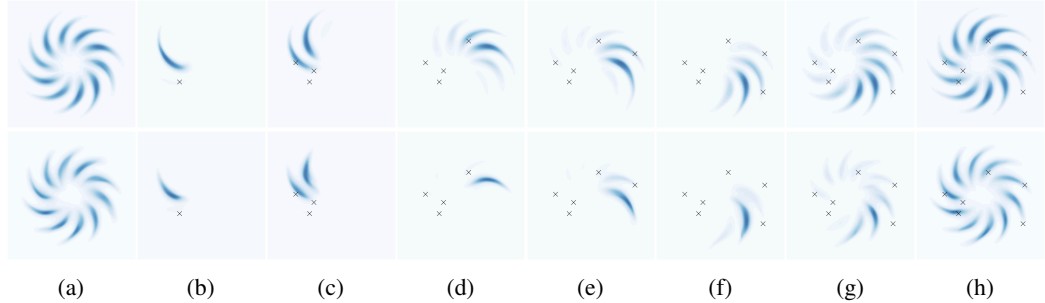

|  |  |  |  |  |  |  |  |
| (a) | (b) | (c) | (d) | (e) | (f) | (g) | (h) |

Figure 5: Evolution of spatial densities on PINWHEEL data. *top:* Attentive CNF. *bottom:* Jump CNF. (a) Before observing any events at $t=0$, the distribution is even across all clusters. (b-f) Each event increases the probability of observing a future event from the subsequent cluster in clock-wise ordering. (g-h) After a period of no new events, the distribution smoothly returns back to the initial distribution (a).

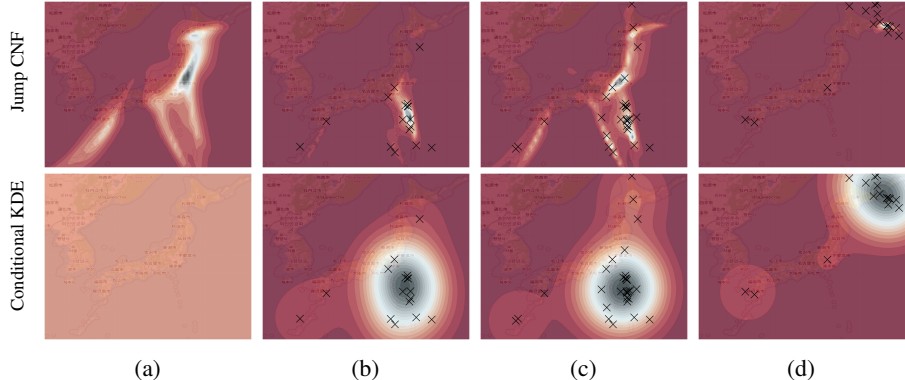

|  |  |  |  |
| (a) | (b) | (c) | (d) |

Figure 7: Snapshots of conditional spatial distributions modeled by the Jump CNF (*top*) and a conditional kernel density estimator (KDE; *bottom*). (a) Distribution before any events at $t=0$. (b-d) The Jump CNF's distributions concentrate around tectonic plate boundaries where earthquakes and aftershocks gather, whereas the KDE must use a large variance in order to capture propagation of aftershocks in multiple directions.

COVID-19 CASES We use data released publicly by The New York Times (2020) on daily COVID-19 cases in New Jersey state. The data is aggregated at the county level, which we dequantize uniformly across the county. Number of events per sequences ranges between 3 to 323.

BOLD5000 This consists of fMRI scans as participants are given visual stimuli (Chang et al., 2019). We convert brain responses into spatio-temporal events following the z-score thresholding approach in Tagliazucchi et al. (2012; 2016). Number of events per sequences ranges between 6 to 1741.

In addition to these datasets, we also report in Appendix A results for CITIBIKE, a data set consisting of rental events from a bike sharing service in New York City.

**Baselines** To evaluate the capability of our proposed models, we compare against commonly-used baselines and state-of-the-art models. In some settings, ground intensity and conditional mark density are independent of each other and we can freely combine different baselines for the temporal and spatial domains. As temporal baselines, we use a homogeneous Poisson process, a self-correction process, a Hawkes process, and the Neural Hawkes Process, which were trained using their officially released code. As spatial baselines, we use a conditional kernel density estimator (KDE) with learned parameters, where $p(\boldsymbol{x}|t)$ is essentially modeled as a history-dependent Gaussian mixture model (see Appendix B), as well as the Time-varying CNF. In addition, we also compare to our implementation of Neural Jump SDEs (Jia & Benson, 2019) where the spatial distribution is a Gaussian mixture model. We use the same architecture as our GRU-based continuous-time hidden states for fair comparison, as we found the simpler parameterization in Jia & Benson (2019) to be numerically unstable for large number of events. Range of hyperparameter values are outlined in Appendix D.

| | Pinwheel | | Earthquakes JP | | COVID-19 NJ | | BOLD5000 | |
|---|---|---|---|---|---|---|---|---|
| Model | Temporal | Spatial | Temporal | Spatial | Temporal | Spatial | Temporal | Spatial |
| Poisson Process | $-0.784_{\pm0.001}$ | – | $-0.111_{\pm0.001}$ | – | $0.878_{\pm0.016}$ | – | $0.862_{\pm0.018}$ | – |
| Self-correcting Process | $-2.117_{\pm0.222}$ | – | $-7.051_{\pm0.780}$ | – | $-10.053_{\pm1.150}$ | – | $-6.470_{\pm0.827}$ | – |
| Hawkes Process | $-0.276_{\pm0.033}$ | – | $0.114_{\pm0.005}$ | – | $2.092_{\pm0.023}$ | – | $2.860_{\pm0.050}$ | – |
| Neural Hawkes Process | $-0.023_{\pm0.001}$ | – | $0.198_{\pm0.001}$ | – | $2.229_{\pm0.013}$ | – | $3.080_{\pm0.019}$ | – |
| Conditional KDE | – | $-2.958_{\pm0.000}$ | – | $-2.259_{\pm0.001}$ | – | $-2.583_{\pm0.000}$ | – | $-3.467_{\pm0.000}$ |
| Time-varying CNF | – | $-2.185_{\pm0.003}$ | – | $-1.459_{\pm0.016}$ | – | $-2.002_{\pm0.002}$ | – | $-1.846_{\pm0.019}$ |
| Neural Jump SDE (GRU) | $-0.006_{\pm0.042}$ | $-2.077_{\pm0.026}$ | $0.186_{\pm0.005}$ | $-1.652_{\pm0.012}$ | $2.251_{\pm0.004}$ | $-2.214_{\pm0.005}$ | $5.675_{\pm0.003}$ | $0.743_{\pm0.089}$ |
| Jump CNF | $0.027_{\pm0.002}$ | $-1.562_{\pm0.015}$ | $0.166_{\pm0.001}$ | $-1.007_{\pm0.050}$ | $2.242_{\pm0.002}$ | $-1.904_{\pm0.004}$ | $5.536_{\pm0.016}$ | $1.246_{\pm0.185}$ |
| Attentive CNF | $0.034_{\pm0.001}$ | $-1.572_{\pm0.002}$ | $0.204_{\pm0.001}$ | $-1.237_{\pm0.075}$ | $2.258_{\pm0.002}$ | $-1.864_{\pm0.001}$ | $5.842_{\pm0.005}$ | $1.252_{\pm0.026}$ |

Table 1: Log-likelihood per event on held-out test data (higher is better). Standard devs. estimated over 3 runs.

**Results & Analyses**    The results of our evaluation are shown in table 1. We highlight all results where the intervals containing one standard deviation away from the mean overlap.

Across all data sets, the Time-varying CNF outperforms the conditional KDE baseline despite not being conditional on history. This suggests that the overall spatial distribution is rather complex. We also see from Figure 7 that Gaussian clusters tend to compensate for far-reaching events by learning a larger band-width whereas a flexible CNF can easily model multi-modal event propagation.

The Jump and Attentive CNF models achieve better log-likelihoods than the Time-varying CNF, suggesting prediction in these data sets benefit from modeling dependence on event history.

For COVID-19, the self-exciting Hawkes process is a strong baseline which aligns with similar results for other infectious diseases (Park et al., 2019), but Neural STPPs can achieve substantially better spatial likelihoods. Overall, NHP is competitive with the Neural Jump SDE; however, it tends to fall short of the Attentive CNF which jointly models spatial and temporal variables.

In a closer comparison to the temporal likelihood of Neural Jump SDEs (Jia & Benson, 2019), we find that overly-restricted spatial models can negatively affect the temporal model since both domains are tightly coupled. Since our realization of Neural Jump SDEs and our STPPs use the same underlying architecture to model the temporal domain, the temporal likelihood values are often close. However, there is still a statistically significant difference between our Neural STPP models and Neural Jump SDEs even for the temporal log-likelihood on all data sets.

Finally, we note that the results of the Jump and Attentive CNFs are typically close. The attentive model generally achieves better temporal log-likelihoods while maintaining competitive spatial log-likelihoods. This difference is likely due to the Attentive CNF's ability to attend to all previous events, while the Jump CNF has to compress all history information inside the hidden state at the time of event. The Attentive CNF also enjoys substantially faster computations (see Appendix A).

## 6    CONCLUSION

To learn high-fidelity models of stochastic events occurring in continuous space and time, we have proposed a new class of parameterizations for spatio-temporal point processes. Our approach combines ideas of Neural Jump SDEs with Continuous Normalizing Flows and allows to retain the flexibility of neural temporal point processes while enabling highly expressive models of continuous marks. We leverage Neural ODEs as a computational method that allows computing, up to negligible numerical error, the likelihood of the joint model, and we show that our approach achieves state-of-the-art performance on spatio-temporal datasets collected from a wide range of domains.

A promising area for future work are applications of our method in earth and climate science which often are concerned with modeling highly complex spatio-temporal data. In this context, the use of Riemannian CNFs (Mathieu & Nickel, 2020; Lou et al., 2020; Falorsi & Forré, 2020) is especially interesting as it allows us to model Neural STPPs on manifolds (*e.g.* the earth's surface) by simply replacing the CNF in our models with a Riemannian counterpart.

ACKNOWLEDGMENTS

We acknowledge the Python community (Van Rossum & Drake Jr, 1995; Oliphant, 2007) for developing the core set of tools that enabled this work, including PyTorch (Paszke et al., 2019), torchdiffeq (Chen, 2018), fairseq (Ott et al., 2019), Jupyter (Kluyver et al., 2016), Matplotlib (Hunter, 2007), seaborn (Waskom et al., 2018), Cython (Behnel et al., 2011), numpy (Oliphant, 2006; Van Der Walt et al., 2011), pandas (McKinney, 2012), and SciPy (Jones et al., 2014).

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

## A   ADDITIONAL RESULTS AND FIGURES

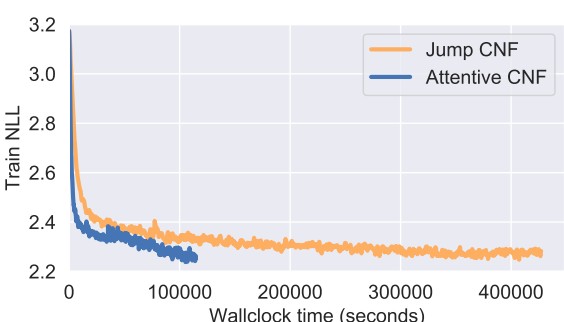

Figure 8: Runtime comparison of Jump and Attentive CNF.

| Model | Citibike NY | |
| --- | --- | --- |
| | Temporal | Spatial |
| Poisson Process | $0.609_{\pm 0.012}$ | – |
| Self-correcting Process | $-5.649_{\pm 1.433}$ | – |
| Hawkes Process | $1.062_{\pm 0.000}$ | – |
| Neural Hawkes Process | $1.030_{\pm 0.015}$ | – |
| Conditional KDE | – | $-2.856_{\pm 0.000}$ |
| Time-varying CNF | – | $-2.132_{\pm 0.012}$ |
| Neural Jump SDE | $1.092_{\pm 0.002}$ | $-2.731_{\pm 0.001}$ |
| Jump CNF | $1.105_{\pm 0.002}$ | $-2.155_{\pm 0.015}$ |
| Attentive CNF | $1.112_{\pm 0.002}$ | $-2.095_{\pm 0.006}$ |

Table 2: Log-likelihood values on held-out test data for an urban mobility data set.

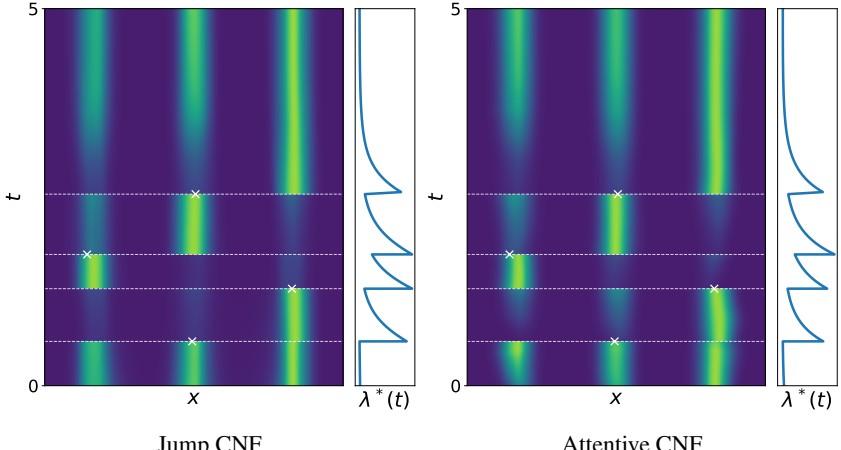

Figure 9: Both the Jump CNF and Attentive CNF are capable of modeling different the spatial distributions based on event history, so the appearance of a new event effectively shifts the distribution instantaneously. Shown on a synthetic 1-D data set similar to PINWHEEL, except we use a mixture of three Gaussians. Each event increases the likelihood of events for the cluster to the right.

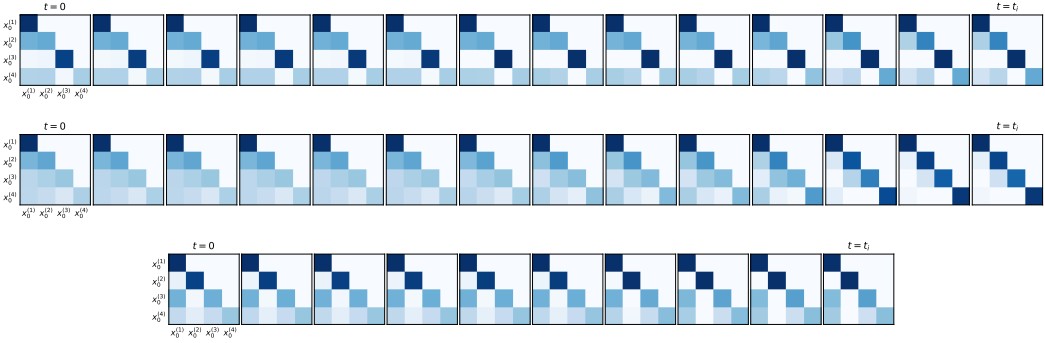

Figure 10: Learned attention weights for random event sequences.

## B    BASELINE

Our self-excitation baseline uses a Hawkes process to model the temporal variable, then uses a Gaussian mixture model to describe the spatial distribution conditioned on history of events. This corresponds to the following likelihood decomposition

$$\log p(t_1, \ldots, t_n, x_1, \ldots, x_n) = \sum_{i=1}^{n} \log p(x_i|t_i, t_1, \ldots, t_{i-1}, x_1, \ldots, x_{i-1}) + \sum_{i=1}^{n} \log p(t_i|t_1, \ldots, t_{i-1})$$

(20)

Note that $t_i$ does not depend on the spatial variables associated with previous events. This dependence structure allows the usage of simple temporal point processes to model $t_i$, *e.g.* a Hawkes process, since temporal variables do not depend on the spatial information. The spatial distribution conditions all past events as well as the current time of occurance.

Our baseline model assumes a simple Gaussian conditional model, that new events are likely to appear near previous events.

$$\log p(x_i|t_i, t_1, \ldots, t_{i-1}, x_1, \ldots, x_{i-1}) = \sum_{j=1}^{i-1} \alpha_j \mathcal{N}(x_j|\sigma^2), \quad \alpha_j = \frac{\exp\{(t_j - t_i)/\tau\}}{\sum_{j'=1}^{i-1} \exp\{(t_{j'} - t_i)/\tau\}}$$

(21)

This parameteric model has two learnable parameters: $\sigma^2$ and $\tau$, which control the rate of decay in the spatial and temporal domains, respectively.

However, this Gaussian spatial model assumes events are propagated in all directions equally and can only model local self-excitation behavior. These assumptions are often used for simplifications but are generally incorrect for many spatio-temporal data. To name a few, earthquakes occur more frequently along boundaries of tectonic plates, epidemics propagate along traffic routes, taxi demands saturate locally and change as customers move around.

## C    PRE-PROCESSING STEPS FOR EACH DATA SET

PINWHEEL  We sample from a multivariate Hawkes process with 10 dimensions. We turn this into continuous spatial variables by assigning each dimension to a cluster from a "pinwheel" distribution, and sample from the corresponding cluster for each event. Number of events per sequences ranges between 4 to 108.

EARTHQUAKES  For modeling earthquakes and aftershocks, we gathered location and time of all earthquakes in Japan from 1990 to 2020 with magnitude of at least 2.5 from the U.S. Geological Survey (2020). Starting from January 01, 1990, we created sequences with a gap of 7 days. Each sequence was of length 30 days. We ensured there was no contamination between train/val/test sets by removing intermediate sequences. We removed earthquakes from 2010 November to 2011 December, as these sequences were too long and only served as outliers in the data. This resulted in 950 training sequences, 50 validation sequences, and 50 test sequences. Number of events per sequence ranges between 18 to 543.

COVID-19 CASES  We use data released publicly by The New York Times (2020) on daily COVID-19 cases in the New Jersey state, from March to July of 2020. The data is aggregated at the county level, which we dequantize uniformly across the county. We also dequantize the temporal axis by assigning new cases uniformly within the day. Starting at March 15, and every 3 days, we took a 7 day length sequence. For each sequence, we sampled each event with a probability of 0.01. This was done 50 times per sequence. We ensured there was no contamination between train/val/test sets by removing intermediate sequences. This resulted in 1450 training sequences, 100 validation sequences, and 100 test sequences. Number of events per sequence ranges between 3 to 323.

CITIBIKE  Citibike is a bike sharing service in New York City. We treat the start of each trip as an event, and use the data from April to August of 2019. We split into sequences of length 1 day starting at 5:00am of each day. For each sequence, we subsampled with a probability of 0.005 per event, 20 times. This resulted in 2440 training sequences, 300 validation sequences, and 320 test sequences. Number of events per sequence ranges between 9 to 231.

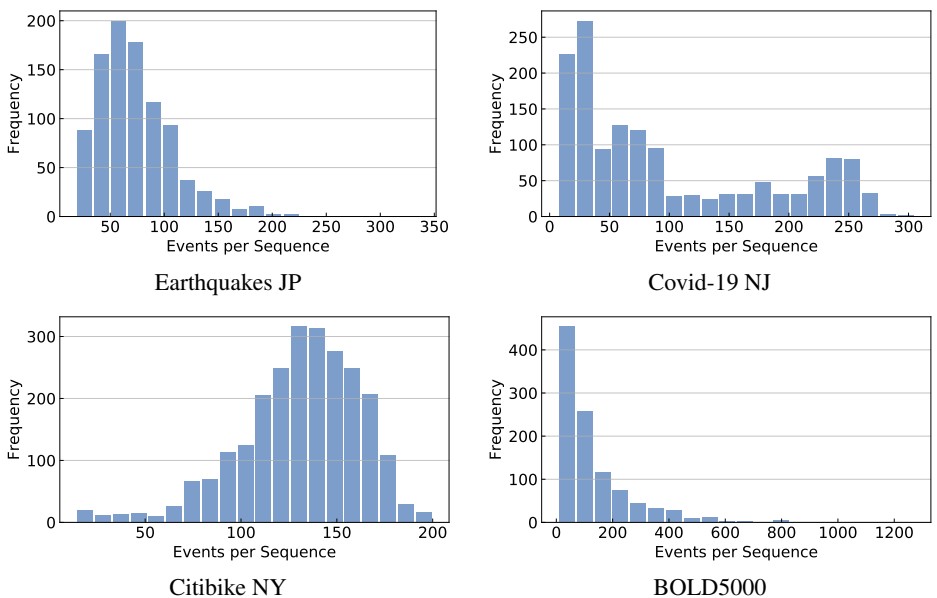

Figure 11: Histograms of the number of events per sequence in each processed data set.

BOLD5000 This consists of fMRI scans of four participants as they are given visual stimuli (Chang et al., 2019). We use the sessions of a single patient and for each run, we split into 3 sequences, treated individually. We converted brain responses into spatio-temporal events following the z-score thresholding approach in Tagliazucchi et al. 2016, Equation (2). We used a threshold of $\gamma = 6.0$. We split the data into 1050 training sequences, 150 val sequences, 220 test sequences. Number of events per sequence ranges between 6 to 1741.

Each data set contains sequences with highly variable number of events, with varying degrees of dependence between events, making them difficult to model with traditional point process models. We plot histograms showing the number of events per sequence in Figure 11.

## D   HYPERPARAMETERS CHOSEN AND TESTED

For the time-varying, jump, and attentive CNF models, we parameterized the CNF drift as a multilayer perceptron (MLP) with dimensions $[d - 64 - 64 - 64 - d]$, where $d$ is the number of spatial variables. We swept over activation functions between using softplus or a time-dependent Swish (Ramachandran et al., 2017).

$$\text{TimeDependentSwish}(t, z) = h\sigma(\beta(t) \odot z) \qquad (22)$$

where $\sigma$ is the logistic sigmoid function, $\odot$ is the Hadamard (element-wise) product, and $\beta : \mathbb{R} \to \mathbb{R}_{d_z}$ is a MLP with widths $[1 - 64 - d_z]$ where $d_z$ is the dimension of $z$, using the softplus activation function. We ultimately decided on using the time-dependent Swish for all experiments.

We swept over the MLP for defining $f_h$ for the continuous-time hidden state in eq. (11) using hidden widths of $[8 - 20]$, $[32 - 32]$, $[64 - 64]$, $[32 - 32 - 32]$, and $[64 - 64 - 64]$. The majority of models used $32 - 32$ as it provided enough flexibility while remaining easy to solve. We used the softplus activation function. We tried MLP for parameterizing the instantaneous change in eq. (12); however, it was too unstable for long sequences. We therefore switched to the GRU parameterization, which takes an input (new event), the hidden state at the time of event, and outputs a new hidden state.

We regularized the $L_2$ norm of the hidden state drift with a strength of 1e-4, chosen from $\{0, 1e-4, 1e-3, 1e-2\}$. We optionally used optimal transport-inspired regularization from Finlay et al. (2020), which adds a Frobenius norm regularization to the gradient of the drift in addition to the $L_2$ norm regularization, to the CNF models with a strength of $\{0, 1e-4, 1e-3, 1e-2\}$. The Time-varying and Attentive CNF models did not require regularization and were mostly kept at 0, but the Jump CNF models benefited from some amount of regularization to avoid numerical instability.

To model a non-trivial spatial distribution for the entire data interval, we shift the data interval to start at $t = 2$ for all CNF models. Thus the interval used for parameterizing the CNF is $[2, T + 2]$. Generally, the "time" variable is a dummy one; we can place the base distribution at any time, and we can choose any interval on the real line to be the data interval; this does not limit the model in any way.

For the Jump CNF, we used a composition of 4 radial flows (Rezende & Mohamed, 2015) to parameterize the instantaneous updates in eq. (17). All parameters of the radial flows were parameterized to be the output of a MLP that takes as input the hidden state at the time of the event (before the hidden state is updated based on the current event). The radial flows were initialized in such a way that the log determinant is near zero.

For the Attentive CNF, the drift function consists of

Time-dependent MLP($d - 64 - 64$) $\rightarrow 2\times$MultiheadAttention $\rightarrow$ Time-dependent MLP($64 - 64 - d$)

where the Time-dependent MLPs make use of the TimeDependentSwish. As was done in Vaswani et al. (2017), the multihead attention is used within a residual branch, except we swapped LayerNorm (Ba et al., 2016) with ActNorm (Kingma & Dhariwal, 2018) as LayerNorm has an unbounded Lipschitz and can be ill-suited for use in ODEs. We tested both standard multihead attention (Vaswani et al., 2017) and the Lipschitz multihead attention (Kim et al., 2020). The Lipschitz multihead attention typically produced similar validation NLL as the standard multihead attention but were more stable on multiple occassions. We therefore kept the Lipschitz multihead attention for all experiments. We additionally, use an auxiliary (non-attentive, simply with the two multihead attention layers removed) CNF to map from 0 (*i.e.* the time of base distribution) to the beginning of the data interval.

We initialized all Neural ODEs (for the hidden state and CNFs) with zero drift by initializing the weights and biases of the final layer to zero.

The log-likelihood values reported are after the spatial variables have been standardized using the empirical mean and standard deviation from the training set.

We train and test on log-likelihood (in nats) per event, which normalizes eq. (8) of each sequence by the number of events.

All integrals were solved using Chen (2018) to within a relative and absolute tolerance of 1E-4 or 1E-6, chosen based on preliminary testing for convergence and stability.

Our implementation of the Neural Jump SDE shares the same continuous-time hidden state parameterization but uses a mixture of Gaussians as the spatial model. We used 5 mixtures, and a MLP that maps from the hidden state to the parameters of this mixture of Gaussians (the means, log standard deviations, and mixture coefficients).

## E  REMOVING CROSS-EVENT PARTIAL DERIVATIVES

This results in a lower-variance gradient estimator for training, and allows parallel computation of conditional log probabilities at test time.

We first summarily describe the attention mechanism. For an input $X \in \mathbb{R}^{n \times d}$ representing the sequence of $n$ variables $\{x_s^{(0)}, \ldots, x_s^{(n)}\}$ at some value of $s$, this attention mechanism creates logits $P \in \mathbb{R}^{n \times n}$ and values $V \in R^{n \times d}$, both dependent on $X$, such that the output is

$$O = \underbrace{\text{softmax}(P)}_{:=S} V. \tag{23}$$

where the softmax is taken over each row of $P$. The output is then added to $X$ as a residual connection. The multihead attention computes $P$ in a way such that $P_{ij}$ depends on $X_i$ and $X_j$, and $V_i$ depends on $X_i$. This is true for both the vanilla MHA (Vaswani et al., 2017) and the L2 MHA (Kim et al., 2020). For our use case, $P_{ij}$ is set to $-\inf$ for $j > i$ as we don't want to attend to future events.

We retain only the block-diagonal gradients where each block contains variables corresponding to one event. This is equivalent to removing all the cross-event dependencies.

$$\frac{\partial O_i}{\partial X_i} = S_{:,i} \frac{\partial V_i}{\partial X_i} + V^\mathsf{T} \frac{\partial S}{\partial P_i} \frac{\partial P_i}{\partial X_i} + V^\mathsf{T} \frac{\partial S}{\partial P_{:,i}} \frac{\partial P_{:,i}}{\partial X_i} \tag{24}$$

# F  PARALLEL SOLVING OF MULTIPLE ODEs WITH VARYING INTERVALS

Our numerical ODE solvers integrate a single ODE system $\frac{dx}{dt} = f(t,x)$, where $x \in \mathbb{R}^d$ and $f : \mathbb{R}^{1+d} \to \mathbb{R}^d$, on a single fixed interval $[t_{start}, t_{end}]$. We can express the inputs and outputs of an ODE solver with

$$\texttt{ODESolve}(x_0, f, t_{start}, t_{end}) \triangleq x_0 + \int_{t_{start}}^{t_{end}} f(t, x(t)) \, dt = x(t_{end}). \tag{25}$$

where $x_0$ is a vector containing the initial state at the initial time $t_0$.

**Multiple ODEs**  Now suppose we have a set of $M$ systems (*i.e.* $\frac{dx_m}{dt} = f_m$ for $m = 1, \ldots, M$) that we would like to solve. If all systems had the same initial time and desired output time of $t_{start}$ and $t_{end}$, we can readily create a joint system

$$x_{joint} = \begin{bmatrix} x_1 \\ \vdots \\ x_M \end{bmatrix} \qquad \text{that follows} \qquad \frac{dx_{joint}}{dt} = \begin{bmatrix} f_1(t, x_1) \\ \vdots \\ f_M(t, x_M) \end{bmatrix} \tag{26}$$

Solving this joint system can be done in parallel with a single call to $\texttt{ODESolve}$:

$$x_{joint}(t_1) = \texttt{ODESolve}(x_{0joint}, f_{joint}, t_0, t_1) \tag{27}$$

which computes $x_m(t_1)$ for all $m = 1, \ldots, M$. This is the standard method used for solving a batch of Neural ODEs.

**Adding dependencies is straightforward**  Note that extending this further, the ODE systems do not need to be independent. They can depend on other variables *at the same concurrent time value* because the joint system is still an *ordinary* differential equation. For instance, we can have

$$\frac{dx_m}{dt} = f_m(t, x_1, \ldots, x_m) \tag{28}$$

where $f_m : \mathbb{R}^{1+Md} \to \mathbb{R}^d$. Each system is now a partial differential equation, but the joint system $x_{joint}$ is still an ODE and can be solved with one call to $\texttt{ODESolve}$.

**Varied time intervals**  Now suppose each system has a different time interval that we want to solve. Different initial times *and* different end times. Let's denote the start and end time for the $m$-th system as $t_{start}^{(m)}$ and $t_{end}^{(m)}$ respectively. We can construct a dummy variable that always integrates from 0 to 1, and perform a change of variables (reparameterization) to transform every system to use this dummy variable.

As a concrete example of this reparameterzation procedure, consider just one system $x(t)$ with drift function $f(t, x)$ that we want to integrate from $t_{start}$ to $t_{end}$ with the initial value $x_0$. We can transform $x(t)$ using the relation $s = \frac{t - t_{start}}{t_{end} - t_{start}}$, or equivalently

$$t = s(t_{end} - t_{start}) + t_{start}, \tag{29}$$

into a solution $\tilde{x}(s)$ on the unit interval $[0, 1]$ such that

$$\tilde{x}(s) = x(s(t_{end} - t_{start}) + t_{start}) \tag{30}$$

The drift function for $\tilde{x}$ then follows as

$$\begin{aligned}
\tilde{f}(s, \tilde{x}(s)) \triangleq \frac{d\tilde{x}(s)}{ds} &= \frac{dx(t)}{dt}\bigg|_{t=s(t_{end}-t_{start})+t_{start}} \frac{dt}{ds} \\
&= f(t, x(t))\bigg|_{t=s(t_{end}-t_{start})+t_{start}} (t_{end} - t_{start}) \\
&= f(s(t_{end} - t_{start}) + t_{start}, \tilde{x}(s))(t_{end} - t_{start})
\end{aligned} \tag{31}$$

Now since $\tilde{x}(0) = x(t_{start})$ and $\tilde{x}(1) = x(t_{end})$, the following are equivalent

$$x(t_{end}) = \texttt{ODESolve}(x_0, \tilde{f}, 0, 1) = \texttt{ODESolve}(x_0, f, t_{start}, t_{end}) \tag{32}$$

**Putting it all together** Let $\tilde{x}_m(s)$ be the reparameterized solution for $x_m(t)$ such that

$$\tilde{x}_m(s) = x_m \left( s \left( t_{end}^{(m)} - t_{start}^{(m)} \right) + t_{start} \right) \tag{33}$$

We can then solve for all $M$ systems, with different varying time intervals, using

$$\tilde{x}_{joint} = \begin{bmatrix} \tilde{x}_1 \\ \vdots \\ \tilde{x}_M \end{bmatrix} \qquad \text{that follows} \qquad \frac{d\tilde{x}_{joint}}{ds} = \begin{bmatrix} \tilde{f}_1(s, \tilde{x}_1) \\ \vdots \\ \tilde{f}_M(s, \tilde{x}_M) \end{bmatrix}. \tag{34}$$

Solving this system to $s = 1$ yields $\tilde{x}_m(1) = x_m(t_{end}^{(m)})$.

Assuming $t_{start}^{(m)} = 0$ for all $m = 1, \ldots, M$ in order to reduce notational complexity, we can write this joint system in terms of the original systems as

$$\frac{d\tilde{x}_{joint}}{ds} = \begin{bmatrix} f_1 \left( st_{end}^{(1)}, \tilde{x}(s) \right) \; t_{end} \\ \vdots \\ f_1 \left( st_{end}^{(1)}, \tilde{x}(s) \right) \; t_{end} \end{bmatrix}. \tag{35}$$

This is the joint system written in equation 14. The joint system in equation 19 adds dependence between the $M$ systems but can still be solved with a single `ODESolve`.

