# OpenReview forum: "Neural Spatio-Temporal Point Processes"
_ICLR.cc/2021/Conference — ICLR 2021 Poster_

### Official Review · AnonReviewer2 · 2020-10-27
**Overall this is a good paper, despite some minor flaws.**

**Rating:** 7
**Confidence:** 4

**Review:**

This paper proposes a novel deep approach to the learning of spatio-temporal point processes via normalizing flows. Overall I think this is a good paper, presenting many interesting ideas and may impact further research on point processes.  The combination of flow-based network structure and the probabilistic model--point process should make sense. The formulation and presentation are good, which makes the paper easy to follow. However, there are a couple of questions for the authors to further address:
1. It seems that the proposed model contains a jump CNF for the mark probability $p^*(\boldsymbol{x}_t|t)$. I'm not sure if it really makes sense that the probability of mark has a "jump" over time. Here such formulation seems to be problematic. It is straightforward for the ground intensity function to consider the discontinuity at the point when an event occurs, as it represents a (self-exiting/inhibitive) temporal point process. The features are often assumed to be homogenous over time. It would be better if the authors can justify such a formulation.
2. It seems intractable for the model to predict the next event. To compute the arrival time and mark for the next event, one should consider integrals with respect to $\lambda$ and $p$, which looks quite complex when they involve ODEs.
3. The authors are advised to further illustrate the attentive CNF. This seems to be a very interesting topic, but the authors only give a brief introduction in very short content. It is not very clear how the authors incorporate the attention mechanism to CNF.
4. The experiment looks somehow weak.  First, the authors criticize that KDE has a large entropy (variance?). However, the variance of KDE depends on the kernel bandwidth, it is not fair to judge based on just one prefixed kernel bandwidth. Second, the authors seem to miss a couple of baselines that deal with the same task in the experiment. The NHP and the RMTPP are also able to model the spatial-temporal point process if the losses are change to the metrics on Euclidean space. Besides, please consider
Li, L., & Zha, H. (2014). Learning parametric models for social infectivity in multi-dimensional Hawkes processes. AAAI 2014.
Li, T., & Ke, Y. (2020). Tweedie-Hawkes Processes: Interpreting the Phenomena of Outbreaks. AAAI 2020.

---

> ### Author Response · Authors · 2020-11-19
> **Thank you for the comments**
>
> We thank the reviewer for these comments. We address reviewer questions below:
>
> > I'm not sure if it really makes sense that the probability of mark has a "jump" over time.
>
> The perceived “jumps” in the distribution are really just due to different event histories. A good way to think about this may be: the spatial distribution at any time value should be different depending on the event history. For instance, an earthquake occurring on the north of Japan will result in aftershocks closer to the north, while an earthquake occurring in the south will result in aftershocks in the south. We use the term “jump” as it is easier to describe constructively how the Jump CNF processes event history: instead of building a new CNF model from scratch, it uses the knowledge that subsequent event histories only add new events.
>
> Note that these jumps in distribution also occur for baseline models, such as a model with self-exciting intensity function. For instance, the models discussed in [1] place a mass on every event in the history, so with a new event, this introduces a discontinuous change in the density centered at that new event location. We included this type of self-exciting baseline as the “conditional KDE” baseline. We do note that our data sets are constructed with strong dependence on event history in mind, as can be seen from the difference in performance between the Time-varying CNF and the history-dependent models.
>
> In the case where the underlying spatial distribution is truly homogeneous, our models can also recover this as a special case, though we agree there may be extra optimization issues and a homogeneous model would be easier to train. Note that we actually initialized our models in such a way that it initially does not make use of jumps. Briefly, the Jump CNF’s instantaneous update is x + phi(x), where phi(x) is zero, and the Attentive CNF’s attention mechanism is initialized with zeros. This point has been further explained in the new appendix section detailing hyperparameters.
>
> > It seems intractable for the model to predict the next event. To compute the arrival time and mark for the next event, one should consider integrals with respect to $\lambda$ and $p$, which looks quite complex when they involve ODEs.
>
> If the reviewer means density computation, the key insight is that by using a CNF, we turn the multivariate integral in equation (2) into a 1d integral in equation (8). This is because the family of normalizing flow models self-normalizes (i.e. always has an integral of one). Furthermore, by making use of continuous-time normalizing flows, we can describe an infinite set of distributions on the real line by simply solving (numerically) the 1d integral in equation (5). All such 1d integrals are solved with an ODE solver to within numerical tolerance. We have also clarified this point in the appendix.
>
> If the reviewer means sampling, then an event handling ODE solver can be used with the inverse sampling approach to sample the time of the next event, and we sample from the CNF by sampling x_0 from the base distribution and solving equation (4). Both likelihood computation and sampling rely on using ODE solvers to compute 1d integrals. We can include an algorithm box in our next revision to make this clearer.
>
> > The authors are advised to further illustrate the attentive CNF. This seems to be a very interesting topic, but the authors only give a brief introduction in very short content. It is not very clear how the authors incorporate the attention mechanism to CNF.
>
> We agree. We have added illustrations in the main text (Figure 2) as well as the learned spatial densities on a 1D data set (Figure 9). A 1d spatio-temporal point process can be visualized in a single figure. We have also added additional experiments (showing the faster convergence of the low-variance estimator) in Figure 3 and additional explanations (on how the integrals can be solved in parallel) in Appendix F.
>
> [1] “A Review of Self-Exciting Spatio-Temporal Point Processes and Their Applications” Reinhart. (2017).

---

> > ### Author Response · Authors · 2020-11-19
> > **Regarding the last point on experiments**
> >
> > > the authors criticize that KDE has a large entropy (variance?)
> >
> > To clarify, we do not pre-fix the kernel bandwidth but rather optimize it. The reason for the large entropy seems to be that aftershocks can appear in multiple directions, or when there are multiple earthquakes occurring at the same time, the KDE needs to take into account earthquakes that occur far apart from each other, thus resulting in a larger bandwidth being optimal. In contrast, the neural STPP models have higher fidelity and micro capabilities. They have learned that earthquakes propagate roughly along tectonic plate boundaries and thus can focus its density on specific areas instead of propagating, from some centroid, in all directions equally. We have also changed the wording from entropy to variance as hinted by the reviewer, since the key distinction is distance-based and we didn’t actually measure entropy.
> >
> > > the authors seem to miss a couple of baselines that deal with the same task in the experiment. The NHP and the RMTPP are also able to model the spatial-temporal point process if the losses are changed to the metrics on Euclidean space.
> >
> > We agree with the reviewer’s suggestion and have added Neural Hawkes Process as a baseline using the official open-source code. It can be seen that our model as well as Neural Jump SDEs outperform NHPs for temporal prediction on our datasets. As the spatial modeling of NHPs and Neural Jump SDEs are identical, the results of Neural Jump SDE can also serve as a guideline for the spatial performance of NHPs. We thank the reviewer for this suggestion as it further demonstrates the advantages of our approach. Moreover, we have added additional references to temporal point process works in the paper.
> >
> > We’d like to clarify what the reviewer means exactly by “if the losses ... Euclidean space”. If the reviewer means the L2 loss, then we have already considered and ablated against a mixture of Gaussians, which is used in the Neural Jump SDE baseline. This is because the log-likelihood metric generalizes the L2 loss, which equivalently is the log-likelihood of a Gaussian spatial model with fixed variance. The results are that this weaker spatial model not only changes the spatial predictions  but also negatively affects the temporal predictions, as they share the same underlying hidden state. To clarify, our spatial variables are represented and modeled in Euclidean space.
> >
> >
> > > The experiment looks somehow weak.
> >
> > We argue against this in our response to AnonReviewer3, which hopefully helps to address any additional concerns that the reviewer has regarding the experimental evaluation. To clarify our evaluation, we have also added a detailed description of the datasets in the Appendix.  For instance, the total number of random variables considered in a single sequence is relatively large in all datasets, Moreover, the different datasets span very different distributions of sequence lengths: while BOLD5000 follows a heavy-tailed distribution, the Citibike dataset is close to Gaussian. We hope this further illustrates the complexity of the datasets and show the quality of our evaluation.

---

### Official Review · AnonReviewer3 · 2020-10-28
**interesting idea; carefully designed models; weak experiments; weak accept**

**Rating:** 7
**Confidence:** 4

**Review:**

The paper proposes a neural-ODE-based point process for spatio-temporal data. Under the general framework, three particular variants are proposed: they handle data with different characteristics and have different computational efficiency.

Pros:

The idea is interestingly novel.

The proposed model architectures are all carefully thought through: each model component is well-motivated, being supported by convincing justification.

The presentation is clear; some technical parts are enjoyable to read.

The empirical results on log-likelihood comparison look compelling.

Cons:

The experiments are somewhat weak: this is the main reason I didn’t give a higher score.

For temporal comparison, there isn’t any neural baseline model.

There isn’t any prediction accuracy comparison.

Empirical analysis is very limited (maybe cuz of limited experiments conducted).

Questions:

The use of * is really non-standard: in statistics and machine learning, * usually denotes somewhat ground-truth. Can authors think of another notation to omit $H$? Or maybe $H$ can be kept since the single-column format is spacious enough to host long equations?

Eqn-(8): why not $\log p(x | t)$? Is this a typo?

RNN (particularly, LSTM) with continuous-time hidden state was proposed by Mei and Eisner 2017, earlier than the cited Rubanova et al. 2019.
Moreover, the math properties described by eqn-(10--12) also hold for Mei & Eisner 2017.
Can authors appropriately acknowledge these connections?

---

> ### Author Response · Authors · 2020-11-19
> **Thank you for these questions**
>
> We thank the reviewer for the time to carefully review our paper.
>
> We first want to address our experimental setup: why these experiments were chosen and why they are suitable to serve as benchmarks for spatio-temporal point processes.
>
> Firstly, apart from the pinwheel data set, these are all *real data sets used in real applications*. Earthquakes is a common application for point process models due to the non-trivial propagation of aftershocks, and even here we observe significant improvement to spatial modeling compared to previous models using kernel density estimators.. Citibike captures the prediction of customer demand for urban mobility in the most populated city in the US; Covid the spread of epidemics along traffic routes; fMRI scans admittedly are commonly used for more discriminative tasks, but having a generative model of neuronal activity can be used, for instance, for predictive long-term behavior. As our paper is centered around contributions to the modeling aspect, we gathered data sets from a wide variety of domains, all of which showcase the capability of having flexible spatio-temporal point process models. Finally, the pinwheel data set was used as a preliminary means to test our models’ capability in modeling extreme changes in spatial distributions, as almost every new event changes the spatial density to an entirely different region. We included this into the benchmark as it can test the conditional propagation capabilities of our spatial models.
>
> Secondly, what we focus on are *non-trivial conditional signals from previous events*. A possible reason for this perceived weakness might be the dimensions of the data set. Admittedly, the data sets we explore are 2-3 spatial dimensions, a far cry from the “high dimensional” setting of images. But CNFs have already shown promising applications in modeling independent data in prior works, and our methods can also readily handle higher spatial dimensions. Instead of simply increasing spatial dimension, all of the data sets we consider have a widely varying number of events per sequence, from as low as 3 events to 1741 events (the total number of variables is the number of events multiplied by 3-4). The higher range is already in the same order as the number of dimensions for images, while our problem formulation has the added difficulty of modeling an arbitrary number of random variables. Modeling of conditional event information is the core problem that we tackle, and there are associated aspects that need to be considered, such as events possibly occurring at any time value. That the data sets exhibit strong reliance on event history can also be seen from the difference in performance between the time-varying CNF, which does not depend on history, and the history-dependent Neural STPP models.
>
> Finally, all data sets are *from open and publicly available sources*; initial sources and preprocessing are detailed in the appendix. Of course, we also plan to release in due time the processing code as easy-to-access benchmarks for future works.
>
> Based on this discussion, we have added some extra information regarding these data sets into the Appendix (see Figure 11), including the number of sequences and histograms of the number of events per sequence.
>
> We also answer the reviewer’s remaining concerns below:
>
> > For temporal comparison, there isn’t any neural baseline model.
>
> We have added the Neural Hawkes Process (NHP) as a baseline. Though our contributions lie in the spatial domain, having more baselines is not a bad idea. We do find that NHP is generally quite good, on par with our implementation of the Neural Jump SDE on most data sets, but is still shy from our Neural STPP models.
>
> > There isn’t any prediction accuracy comparison.
>
> We only consider continuous marks, and as such, cannot apply an accuracy metric. We do note that discrete marks can be easily added, but they have been explored extensively in previous works and our main focus for the paper is on spatial modeling of continuous variables. We do note that comparing log-likelihood is equivalent to comparing the KL divergence between the target distribution and the model distribution, and it is the main metric used in many probabilistic modeling papers.
>
> > Empirical analysis is very limited (maybe cuz of limited experiments conducted).
>
> We have broken down our empirical analyses into paragraphs that emphasize the distinct comparisons. We compared across almost every pair of models, with key ablations included, e.g. we observed a slight but noticeable gain in temporal log-likelihood when the spatial model is more flexible. Runtime comparison and effect of estimator variance are also included in separate sections.

---

> > ### Author Response · Authors · 2020-11-19
> > **(cont)**
> >
> > For the remaining questions,
> >
> > We simply followed the convention in the point process literature, where the * shorthand is quite common (e.g. see the standard reference Daley & Vere-Jones 2003).
> >
> > We have fixed the typo; thanks for pointing it out.
> >
> > We agree and have added references to other works on continuous-time hidden states, NHP and GRU-decay, when we first discuss this. The key distinction is that they use a linear ODE with an analytical solution, whereas we use the generalization where the ODE is a neural network, which nicely complements the time integral for the CNF component.

---

### Official Review · AnonReviewer5 · 2020-11-05
**review for #665**

**Rating:** 5
**Confidence:** 4

**Review:**

This work investigates a new class of parameterizations for spatio-temporal point processes
which uses Neural ODE to enable flexible, high-fidelity models of discrete events that are localized in continuous time and space.

Strengths: this work is essentially an extension of the Neural Jump SDEs [Jia & Benson (2019)] where the temporal dynamics is modeled as an ODE and the spatial pdf is modeled as a history dependent Gaussian mixture distribution. In this work, the spatial pdf is further extended to an ODE based dynamics. For this purpose, three different continuous normalizing flow models are proposed (Time-Varying CNF, Jump CNF, Attentive CNF). Also, a large number of experiments are conducted and baselines are compared to validate the conclusion.

I recommend rejection at the current stage for the reasons below.

Weakness: A major concern is, if my understanding is right, every mark x^(i) is modeled as an ODE of x^(i)_t on [0, t_i] in the in Time-Varying CNF and Attentive CNF, so there are N (the number of points) ODEs in the model. This setup is problematic because any points except the 1st are impossible to happen at time 0, so they impossibly possess a mark x^(i) at time 0 (in fact, any time before t_(i-1) is impossible). A more reasonable way to characterize the dynamics of x^(i) is to model the ODE on [t_(i-1), t_i] which is used in the Jump CNF. I understand this setup contributes to the parallel computation with the reparameterization trick. In fact, this is reason why both Time-Varying CNF and Attentive CNF can be computed in parallel, but Jump CNF cannot. The Attentive CNF can be seen as a generalized version of Time-Varying CNF due to the introduction of history dependence, but the Jump CNF is a different model as stated above.
Also, the jump CNF can model the abrupt change of the spatial pdf but the Time-Varying CNF and Attentive CNF cannot. Theoretically speaking, the jump CNF should have a more powerful fitting capability (assuming other parts are same) compared with those two models. Why does the Attentive CNF model achieve a better or close performance than jump CNF in most experiments? Does that mean the dynamics in most datasets have no discontinuity? Maybe a simple synthetic experiment with discontinuity in dynamics can help prove this.

Some specific concerns: some synthetic data experiments with specific setup (e.g. discontinuity) are needed to give a deep understanding of the two proposed spatial CNF models.

Typo: or-->of, the second line from the bottom in the first page.
 0-->1, the second line of Eq.(19).

---

> ### Author Response · Authors · 2020-11-19
> **Thank you for the review; we argue that these concerns are not true and have updated the paper to clarify these points**
>
> We thank the reviewer for these questions. If we understand correctly, the reviewer has two main concerns: (i) that we cannot predict the spatial mark at time 0, and (ii) that the Attentive CNF cannot model a discontinuous change to density following new event observations.
>
> We argue that both of these concerns are not true, although the initial submission may not have been sufficiently clear on these points.
>
> To address point (i), we actually set the base distribution to be a couple units of time before the event data interval. This allows the models to have a flexible (non-Gaussian) distribution for predicting the first event. Generally, the t variable is a “dummy” one; we can place the base distribution at any time, and we can choose any interval on the real line to be the data interval; this does not limit the model in any way. This explanation was indeed missing from the paper and we have added this explicitly into the main text. We also note that figures 5 and 7 both contain the spatial distribution before any events occur as the left-most image.
>
> To address point (ii), we did not completely follow the reviewer’s reasoning but we hope the following can answer the reviewer’s concern:
>
> Starting with the Time-varying CNF. This model uses the same drift at every time value for every event. Thus it has the same distribution for all events, regardless of event history, and does not model sudden changes to the distribution based on new events. This we agree with the reviewer.
>
> The Attentive CNF uses different drift functions at every time value for different events (difference in drift is because they have different event histories). Thus, it can model different distributions at the same time value depending on what is in the event history. This allows the model to have a perceived sudden change in distribution once a new event occurs, because the event history pre-event and post-event will be different. We have added visualization on a  1-D data set to aid this explanation. The spatial density is visualized for both Jump and Attentive CNF in Figure 9.
>
> The pinwheel synthetic data set was designed specifically to have sudden changes to spatial distribution, instantaneously moving the mass from one cluster to another (disjoint) cluster. The spatial distributions shown in figure 5 are extremely close together time-wise and the change is due to the addition of new events. It’s admittedly difficult to show this without a video, so instead we added models trained on a 1D data set for visualization.
>
> We hope this correctly addresses the reviewer’s concerns, but if not, please kindly let us know ahead of time before the end of the discussion period.
>
> We also thank the reviewer for raising these concerns, as they have helped us improve the conceptual explanations in the paper.

---

### Official Review · AnonReviewer4 · 2020-11-10
**This paper proposed a continuous space and time process for micro modeling discrete time event sequences. The idea of the paper is very interesting by making point processes leverage the flexibility and tractability of flows in their intensity models.**

**Rating:** 6
**Confidence:** 3

**Review:**

The paper is generally well written.

Section 3 is a little confusing as it's not readily clear which part of the model is using flows. Or what are the flow parameters that needed to be estimated.  How  will the neural network structure look like? How is it trained?

One might wonder why a flow based model is used among these many deep generative models? Where does this invertibility help out?

The GRU model is not well elaborated and is a little unclear.

It would have been interesting to see where the attention model usually attends to. Either in a real world data set or in simple intuitive toy tasks.

Equation 14 and 19 are hard to follow. It's good to elaborate on them. At least in an appendix section due to space limitations.

Will the log-likelihood sufficient for evaluation? Why not more intuitive tasks like event or time predictions? More explanation and justification would have been helpful.

---

> ### Author Response · Authors · 2020-11-19
> **Thank you for these questions**
>
> We thank the reviewer for these questions. As requested, we have updated the paper to include further explanations. Below, we answer their specific questions:
>
> > Section 3 is a little confusing as it's not readily clear which part of the model is using flows. Or what are the flow parameters that needed to be estimated.  How will the neural network structure look like? How is it trained?
>
> We have added to the Appendix more model and training details, detailing the final hyperparameters used (network architecture, optimization) as well as the set of values we experimented with. To summarize for the reviewer: the parameters are all part of either drift functions or instantaneous update functions. The hidden state has a MLP drift function and uses a GRU for the instantaneous update. The Time-varying and Attentive CNFs have only a drift function, while the JumpCNF additionally uses a discrete-time flow for instantaneous updates. Architectures and hyperparameters of these functions are detailed in the Appendix now. Training was done by maximizing the log-likelihood per event, which are equations (5) for Time-varying and Attentive CNF and equation (18) for Jump CNF. All integrals are solved using an ODE solver to within numerical tolerance. We have also added a short tutorial (Appendix F) for how a batch of integrals can be solved efficiently.
>
> > One might wonder why a flow based model is used among these many deep generative models? Where does this invertibility help out?
>
> Apart from the additional flexibility (normalizing flows have been proven to be universal density estimators, which is now cited in the intro), the main reason is tractability. We build on the framework of spatio-temporal point processes, which require solving an integral over the spatial domain (eq 2). By using normalizing flows, we know the normalization constant is one and equation (8) is an immediate consequence of applying this to equation (2). Furthermore, the use of continuous-time normalizing flows (CNF) allows us to model an infinite set of such flexible distributions on the real line by numerically solving just a 1d integral. The use of CNFs also nicely complements the continuous-time parameterization of TPPs, allowing us to numerically solve for both jointly during sampling. In contrast, other deep generative models have a hard time even computing the log-likelihood and have to resort to their own specific means of training.
>
> > The GRU model is not well elaborated and is a little unclear.
>
> We did not elaborate further in the main text as they are quite similar to many previous works. The general idea is that in addition to applying an RNN update at every event time, we also change the hidden state between event times using an ODE. Thus the hidden state changes both at event occurrences (to summarize history) and when no events occur (to predict the immediate future). We have added more references for this continuous-time hidden state in the main text.
>
> > It would have been interesting to see where the attention model usually attends to. Either in a real world data set or in simple intuitive toy tasks.
>
> Some examples have now been added to the Appendix as Figure 10 for a simple toy sequence. We agree that a systematic approach for investigating the interpretability of attentive mechanics in continuous-time could potentially be interesting. However, we believe this is beyond the scope of this paper.
>
> > Equation 14 and 19 are hard to follow. It's good to elaborate on them. At least in an appendix section due to space limitations.
>
> Appendix F now details the derivation for these equations, and we’ve added an intuitive explanation in the main text. We thank the reviewer for this suggestion; there was actually a typo from before.. To summarize, we essentially map each ODE, f, from t in [0, t_i] to an ODE, f_mod, on the interval s in [0, 1], by setting t = s * t_i and post-multiplying the output by t_i (the length of the original interval). This results in f_mod(s) = t_i f(t = s * t_i), which has the same solutions as the original ODE but allows batch solving of multiple ODEs from different intervals. This is more of an engineering consideration to get around our existing ODE solvers being capable of only integrating on a single interval.
>
> > Will the log-likelihood sufficient for evaluation? Why not more intuitive tasks like event or time predictions? More explanation and justification would have been helpful.
>
> We believe that the log-likehood is indeed a good evaluation metric for continuous marks. The Log-likelihood generalizes regression losses (e.g the L2 norm / MSE loss)  for prediction of continuous variables, and comparing log-likelihood is the same as comparing the KL divergence between the target density and the learned model density. We believe it is for these reasons, that log-likelihood is widely used in probabilistic modeling as a metric.

---

### Author Response · Authors · 2020-11-19
**A summary of concerns and updates**

We thank all reviewers for their suggestions and for spending the time to review our paper. We have responded to every reviewer individually. Here, we list the main concerns and how we updated the paper in light of these:

> Experimental details were not completely clear.

We have included all model details, including hyperparameters that were considered during experimentation, into Appendix C.

> General questions regarding the Attentive CNF.

We have included new visualizations based on models trained on a 1-D spatiotemporal data set in Figures 1 and 9.

A new Figure 2 also illustrates how different sequences get treated by the Attentive CNF, showing that it models different spatials distributions when the event history is different. This effectively allows the Attentive CNF to change its distribution instantaneously based on new event observations.

Some sample attention weights are shown in Figure 10, for some short sequences from the 1-D data set.

We have also added an ablation experiment in Figure 3 showing that our low-variance estimator for the log-likelihood leads to faster convergence and better converged models than the naive Hutchinson estimator.

The next point also clears up how Attentive CNFs can be trained efficiently.

> How we efficiently solve multiple non-independent ODEs on different intervals.

We have included a new Appendix F, which contains a tutorial on how we jointly solve multiple (non-independent) ODEs on different time intervals with a single call to an ODE solver. This derives equations (14) and (19) in the paper, and discusses their significance.

> “Experiments are / look weak”.

We took this comment two ways: that it is unclear how meaningful it is to benchmark on these data sets, and that analysis regarding model comparisons were not readily apparent.

For the first point, we have included detailed information regarding these data sets, and have clarified why we chose them. Briefly, because they are all real data sets considered in real applications spanning a wide variety of fields, the event sequences can be quite large and are from heterogeneous distributions (e.g., thousands of random variables, all the while having a large variance in sequence lengths), and they are all gathered from open sources making them suitable for future research.

For the second point, we have broken down our analysis section into simpler points. The main points are that use of high fidelity models is largely beneficial for the considered data sets, better handling of event history is beneficial, and more flexible spatial models also leads to better temporal predictions. We have also included results for the Neural Hawkes Process, which helps us drive the last point.

---

### Decision · Program_Chairs · 2021-01-07
**Final Decision**

**Decision:**

Accept (Poster)

**Comment:**

This paper presents a model for spatiotemporal point processes using neural ODEs. Some technical innovations are introduced to allow the conditional intensity to change discontinuously in response to new events. Likewise, the spatial intensity is expanded upon that proposed in prior work on neural SDEs. Reviewers were generally positive about the contributions and the empirical assessments, and the authors made substantial improvements during the discussion phase.